# Inflammatory Mechanisms in a Neurovascular Disease: Cerebral Cavernous Malformation

**DOI:** 10.3390/brainsci13091336

**Published:** 2023-09-17

**Authors:** Ying Li, Abhinav Srinath, Roberto J. Alcazar-Felix, Stephanie Hage, Akash Bindal, Rhonda Lightle, Robert Shenkar, Changbin Shi, Romuald Girard, Issam A. Awad

**Affiliations:** 1Department of Neurosurgery, First Affiliated Hospital of Harbin Medical University, Harbin 150001, China; yingli.uoc2023@outlook.com (Y.L.); changbinshi@hotmail.com (C.S.); 2Neurovascular Surgery Program, Department of Neurological Surgery, The University of Chicago, Chicago, IL 60637, USA; abhinav.srinath@uchospitals.edu (A.S.); ralcazarf@bsd.uchicago.edu (R.J.A.-F.); shage@bsd.uchicago.edu (S.H.); akbindal@uchicago.edu (A.B.); rlightle@bsd.uchicago.edu (R.L.); rshenkar@uchicago.edu (R.S.); rgirard@bsd.uchicago.edu (R.G.); 3Department of Neurological Surgery, University of Chicago Medicine, 5841 S Maryland, MC3026/Neurosurgery J341, Chicago, IL 60637, USA

**Keywords:** cerebral cavernous malformation, immune response, inflammation, transcriptome, immunothrombosis

## Abstract

Cerebral cavernous malformation (CCM) is a common cerebrovascular malformation causing intracranial hemorrhage, seizures, and focal neurologic deficits. A unique CCM lesional inflammatory microenvironment has been shown to influence the clinical course of the disease. This review addresses the inflammatory cell infiltrate in the CCM lesion and the role of a defined antigen-driven immune response in pathogenicity. We summarize immune mechanisms associated with the loss of the CCM gene and disease progression, including the potential role of immunothrombosis. We also review evidence of circulating inflammatory biomarkers associated with CCM disease and its clinical activity. We articulate future directions for this research, including the role of individual cell type contributions to the immune response in CCM, single cell transcriptomics of inflammatory cells, biomarker development, and therapeutic implications. The concepts are applicable for developing diagnostic and treatment strategies for CCM and for studying other neurovascular diseases.

## 1. Introduction

Cerebral cavernous malformation (CCM), also known as cerebral cavernous angiomas, is a common cerebrovascular malformation with a population prevalence estimated between 0.3 and 0.9% [1]. The lesion consists of dilated capillaries (caverns) with sluggish blood flow, containing thrombus at various stages of organization, in addition to perilesional blood products and a robust inflammatory cell infiltration. Symptomatic clinical presentations include seizures, focal neurologic deficits, and hemorrhagic stroke [2,3]. The CCM can be inherited in an autosomal dominant familial form (20–30% of cases) with multiple lesions throughout the brain or present in a sporadic form with a solitary lesion or lesions clustered around a developmental venous anomaly (70–80% of cases) [1,2]. The identification of genes implicated in CCM development has not fully explained the great variability in lesional clinical behavior [4]. Epidemiological studies have shown that patients can harbor an identical CCM genotype while experiencing different clinical courses of the disease during their lifetime [5,6,7,8,9,10].

Over the past 30 years, immune cell infiltration in CCMs, their interaction with the lesional milieu, and potential pathogenicity have been investigated [11,12,13,14,15,16]. Pro-inflammatory genotypes and gut microbiome factors have been implicated in lesional activity [6,17,18], and the CCM transcriptome studies and circulating biomarkers have endorsed the potential roles of immune response and inflammatory activity in this disease. We here review these concepts and how they influence potential biomarker development and therapy in CCM, as well as how they may be applied to other neurovascular diseases.

### 1.1. Inflammatory Cell Infiltrate in CCM, Predominance of B Cells and Their Clonal Expansion, Potential Pathogenicity

Familial and sporadic CCMs are histopathologically indistinguishable [4,13]. Regardless of genotype or recent clinical activity, resected lesions included a rich infiltration of B-cells and T-cells, human leukocyte antigen-DR-expressing cells, as well as macrophages [13]. As compared to arteriovenous malformations, a preponderance of B cells was documented in CCM and an oligoclonal IgG immune response in lesions [19], including oligoclonality of IgG mRNA [13]. De Souza et al. (2019) reported immune cell infiltration in association with CCM lesion aggressiveness [20]. Other authors have attributed the recurrence of CCM to local inflammatory conditions [21]. 

Later, Shi and co-workers demonstrated in situ B-cell clonal expansion and antigen-driven affinity maturation in CCMs [14]. These inflammatory cell niches are adjacent to abnormal vascular channels, suggesting an organ-intrinsic adaptive immune response in CCMs (Figure 1 and Figure 2).

A preclinical study showed that *Ccm3* mice treated with anti-mouse BR3 to deplete B cells harbored smaller lesions, fewer mature CCMs [15], and decreased non-heme iron deposition, confirming the pathogenicity of the B-cell-mediated immune response in CCMs [15,22]. Additional studies further elucidated that these immune cells target cytoskeletal elements, including vimentin, myosin, and tubulin, that are commonly present in endothelial cells (ECs) and astrocytes [23]. This result also suggests that *in situ* inflammatory cells may react with other lesional cellular components such as neurons, pericytes, and other cell types [23]. 

Neutrophils, which have been identified in CCM, have been described as playing a role in the development of an inflammatory response and induction of a prothrombotic state in other diseases, such as intracerebral hemorrhage (ICH), cerebral arteriovenous malformations, and cancer [24,25,26,27]. The recruitment of neutrophils as well as the deposition of neutrophil extracellular traps (NETs) to neuroinflammatory sites have been shown in CCMs [28]. Nobiletti et al. (2022) demonstrated the role of the CCM1 gene (*KRIT1*) in the regulation of neutrophil adhesion and motility [29]. NET formation, also known as NETosis, is considered the innate immunity function of neutrophils [30], which can be triggered by proinflammatory cytokines as well as activated platelets and ECs [27,31]. Excessive NETosis can result in further endothelial damage due to its cytotoxic effect [28,32]. NETs could mediate adaptive immunity by inducing the secretion of proinflammatory chemokines/cytokines and/or present autoantigens to active B and T cells [33,34]. NET formation and release processes also depend on the generation of reactive oxygen species (ROS), a process shown to play an important role in CCM pathogenesis [35]. While the relationship of NOS to CCM pathogenicity requires further study, a Phase 2 trial targeting ROS to treat symptomatic CCM is ongoing (NCT05085561). 

Macrophages and microglia identified in CCM have well-defined roles in the breakdown of red blood cells as well as heme iron and have been shown to react acutely following an ICH [13,36]. Activated microglia release pro-inflammatory mediators such as interleukin (IL)-1β, IL-6, tumor necrosis factor (TNF)-α, and ROS essential for the innate immune response, leading to neurotoxic sequelae [37,38]. Of interest, pro-inflammatory IL-1β has been suggested to be involved in the recruitment of leukocytes, while IL-6 may be a critical cytokine, controlling the transition from innate to adaptive immunity in the brain [39]. Moreover, TNF-α has been shown to play a role in T cell adhesion to brain ECs, which may occur during neuroinflammation [40]. Activation of microglia and the overproduction of ROS have been reported in the chronic inflammation microenvironment [41]. It contributes to secondary injury cascades through mitogen-activated protein kinase/nuclear factor-κB (MAPK/NF-κB) signaling and NOD-, LRR-, and pyrin domain-containing protein 3 inflammasome activation [41]. 

RhoA/Rho kinase (ROCK) signaling is negatively regulated by CCM proteins, and its dysregulation can increase endothelial permeability [42,43]. In addition, the activation of ROCK not only enhances leukocyte recruitment but also regulates B and T cell activation, proliferation, and cytokine production [44,45]. Lesional B and T cell infiltration was shown to be decreased in CCM preclinical models when blocking the Rho/ROCK signaling pathway [22]. Heterogeneities in the inflammatory cell populations within the lesional micro-environment have been shown to correlate with phenotype characteristics or clinical outcomes [13,46]. Sporadic CCMs harbor greater T-cell infiltration than familial CCMs [13]. Similarly, a higher Th17/Treg ratio has been observed in CCM with clinical symptoms suggestive of a primarily pro-inflammatory T cell phenotype [46].

Other researchers have linked immune responses in CCMs as well as other cerebrovascular anomalies to the lymphatic system and cerebrospinal fluid flow [47]. Low fluid shear stress conditions have been shown to contribute to the activation of cerebral cavernous malformation signaling pathways [48]. They have also been implicated in endothelial inflammation [49,50].

### 1.2. Transcriptomic Studies Highlight Inflammatory and Immune Mechanisms in CCM

The transcriptome of human and murine CCMs has clarified several features of this disease, such as lesion genesis, maturation, growth, and intralesional and extralesional hemorrhage. Some of the mechanistic findings observed in preclinical models of CCM disease were translated and validated in human CCM [16,51,52,53,54]. The transcriptomes of CCM lesional neurovascular units served as a reference to validate mechanistic hypotheses related to the pathogenesis of CCMs [16,55]. Interestingly, common pathways related to immune response, such as leukocyte migration and regulation of antigen receptor-mediated signaling are found across human CCMs, mouse BMECs, and *Caenorhabditis elegans,* as well as in two different genotypes (*Ccm1* and *Ccm3*) [16]. In addition, several pathways related to macrophage cytokine production and activation (i.e., innate immunity) as well as activation and regulation of B and T cells (i.e., adaptive immunity) were identified in both human CCM and mouse BMECs [16]. The transcriptome of in vitro mouse brain microvascular ECs (BMECs) with induced allelic loss of *Ccm3*/*Pdcd10* specifically identified several pathways associated with innate immunity pathways such as macrophage/granulocyte chemotaxis and migration [52].

Of interest, Chapman et al. (2019) reported that a mutation on *kri-1(ok1251)* in the *C. elegans* model affects several genes that are predicted to be involved in innate immunity [54]. These results suggest that a mutation in the *CCM* gene affects innate immunity pathways. Fusco et al. (2022) published transcriptome analyses revealing altered expression of genes involved in hypoxia, inflammation, and immune regulation in *Ccm3* (*Pdcd10*)-depleted mouse endothelial cells [56].

Later, the analyses of the lesional transcriptome of acute and chronic *Ccm3* mouse preclinical models showed that adaptive immune response pathways, such as chemotaxis and activation of neutrophils and leukocytes, chemokine production, and cytokine secretion, were only enriched in the chronic model with more mature lesions akin to symptomatic human CCMs [52]. The transcriptome of the acute *Ccm3* model mostly identified enriched angiogenesis and cell proliferation pathways [52]. This result suggests that chronic bleeding in CCM accompanies an inflammatory stimulus, which may result in a local adaptive immune response. Finally, the transcriptome of surgically resected human CCMs with recent bleeding showed several enriched pathways associated with acute inflammation compared to non-recent bleeding [55]. The CCMs that recently bled likely carry unique biologic signatures, reflecting recent bleeding and subsequent vulnerability to hemorrhage [55].

The contribution of each cell type within the lesional milieu to CCM pathogenesis remains unclear. Pericytes, microglia, astrocytes, and neurons are necessary to maintain the integrity of the blood-brain barrier (BBB) and have also been associated with leukocyte infiltration and the local inflammatory response [12,57,58]. Pericytes normally regulate vascular integrity through close physical contact with ECs as well as paracrine signaling. Brain pericytes have also been shown to have similar functions to immune-regulating cells [59]. Recent preclinical studies on CCM lesional pericytes have identified leukocyte migration/proliferation, leukocyte cell-cell adhesion/activation, and the chemokine signaling pathway [57]. Moreover, pericytes express receptors for DAMPs and inflammatory mediators (IL-1β, IL-6, TNF-α, nitric oxide [NO], matrix metallopeptidase [MMP]-2, and MMP-9) that are associated with inflammatory responses in the brain [60,61]. In response to inflammatory stimuli, pericytes may also have macrophage-like functions such as phagocytosis and act as antigen-presenting cells by displaying antigens through major histocompatibility complex class II [61,62].

### 1.3. Immunothrombosis in CCM

Thrombosis is known to promote an inflammatory response involving a rich repertoire of immune cell infiltration [63,64,65]. The hallmarks of every CCM lesion are the organized thrombi in vascular caverns and leaky endothelium, with blood breakdown products in the surrounding brain parenchyma [11]. A complex interplay occurs between thrombotic and hemostatic regulatory pathways, mediated by cells of the neurovascular unit, and appears to lead to a proinflammatory milieu in brain tissue directly adjacent to the CCM [11,12,66]. A loss of hemostasis due to dysfunctional endothelium has been reported in CCM, resulting in upregulation of both pro- and anti-coagulant factors [51,67]. An upregulation of KLF4, a driver of CCM pathogenesis in mutant ECs, leads to the upregulation of the anticoagulant factors thrombomodulin (TM) and endothelial NO synthase (eNOS) [68]. A local increased level of TM and endothelial protein C receptors generates anticoagulant activated protein C, which could be associated with bleeding in CCMs [51]. 

Procoagulant domains that contribute to inflammation and the breakdown of endothelial integrity have been identified in CCM in conjunction with the inflammatory response induced by anticoagulant lesional domains [51,66]. This prothrombotic environment is a result of both endothelial and non-cell autonomous effects. ECs, as a proinflammatory effector, in the presence of Factor Xa can elevate expression of IL-8, IL-6, monocyte chemotactic protein-1, as well as intercellular adhesion molecule-1 (ICAM-1) and vascular cell adhesion molecule-1 (VCAM-1), which play roles in the promotion of leukocyte adhesion [69]. ECs are also responsible for the production of von Willebrand factor (vWF), which facilitates T-cell and neutrophil recruitment depending on the presence of platelets and the accessibility of their VWF-receptor glycoprotein (GP) Ib-IX-V complex [70]. Given a marked accumulation of vWF in the lumen of CCMs, recruitment of leukocytes can create positive feedback by increasing inflammatory cells around the CCM [11,67].

Perilesional astrocytes have been shown to upregulate coagulant tissue factor production in a *Ccm3* murine model, likely due to increased shear stress caused by diminished actin rearrangement capability in CCM ECs [71,72]. Procoagulant domains and thrombi formation, in addition to low flow through caverns, induce a local hypoxia within the CCM, leading to increased production of endothelial NO and hypoxia-inducible factor (HIF)-1α [11,73]. Increased HIF-1α induces upregulation of VEGF-A in astrocytes, which has been linked to BBB breakdown by targeting endothelial tight junction proteins such as claudin-5 and occludin [73,74]. Astrocytes also play an active role as immune effector cells through the secretion of proinflammatory factors, including TNF-α and IL-1β, at high thrombin levels [75]. Increased levels of TNF-α have been shown to increase expression of ICAM-1 and VCAM-1, increasing trans-endothelial leukocyte migration and further contributing to the proinflammatory environment [76]. Similarly, microglia have shown reactive morphologies in a *Ccm3* murine model within areas colocalized with thrombi and have been positively associated with the number of thrombi around lesions [12]. ICAM-1 stabilization and recruitment of neutrophils may form a feedback loop in which procoagulant-induced inflammation can lead to further endothelium injury and the release of cytokines such as IL-6, IL-8, and TNF-α [77,78,79,80]. These cytokines have been shown to promote the release of vWF from ECs [81,82], allowing for the recruitment of inflammatory cells into the lesional milieu. 

These findings suggest a disrupted balance between pro- and anti-coagulant domains within the CCM to provide a nidus for the recruitment of pro-inflammatory cells and cytokines, further weakening the angio-architecture and leading to neurological sequelae (Figure 3).

### 1.4. Biomarkers and Clinical Features of CCM Disease

The U.S. FDA/NIH have outlined definitions for categorizing biomarkers in relation to their clinical contexts of use [83,84]. Molecular or imaging signatures have been proposed for the diagnosis, etiologic, and severity categorization of several neurological and neuro-vascular disorders [55,85,86,87,88]. The first association between the clinical course of CCM disease and circulating systematic compounds reported a hierarchical cluster of 5 plasma inflammatory cytokines, including IL-2, IFN-γ, TNF-α, IL-1β, and IL-6, defining CCM patients with a “high” inflammatory state, which was associated with seizures and more than one prior SH during a patient’s lifetime [89]. Several translational studies have recently reported that the complex inflammatory and angiogenic milieu observed in the transcriptome of CCMs is reflected in the plasma of CCM patients [16,55]. The lesional transcriptomes have since served as a reference to not only validate mechanistic hypotheses related to the pathogenesis of CCMs but also identify circulating mechanistically plausible candidate biomarkers [16,55]. While many of the biomarkers are not specific to CCM and have been invoked in other inflammatory conditions, they have been linked to the lesional CCM transcriptome and CCM-specific contexts of use. A novel paradigm that integrates multi-omic mechanistic discoveries and applies a likelihood-based computational model was developed to generate and improve biomarker candidates of disease activity [90]. This approach generated two weighted biomarkers based on the combination of plasma levels of inflammatory and angiogenic proteins, able to distinguish CCM patients recent (i.e., diagnostic) and impendent (i.e., prognostic) hemorrhagic activity with up to 80% sensitivity and specificity [55]. There is currently a multi-site initiative to develop highly accurate as well as generalizable diagnostic and prognostic biomarkers of symptomatic hemorrhage in CCMs (R01 NS114552), powered to examine the independent effects of confounders including sex, genotype, age, and lesion location.

The analyses of circulating genetic traits have also shown genetic polymorphisms within pro-inflammatory and immune response genes, such as *TGFBR2*, *CD14*, *IL-6R*, *MSR1*, *IGH*, and *TLR4*, that have been associated with total lesion count, number of large lesions, and intracerebral hemorrhage [6,8,17]. In addition, circulating microRNAs (miRNAs) have been suggested as possible mechanistic biomarkers of hemorrhagic activity for CCM disease. Lyne et al. (2019) identified 13 differently expressed (DE) miRNAs in the plasma of patients who experienced recent CCM-related brain bleeding [55]. Further computational analyses showed that *miR-185-5p*, one of these 13 DE miRNAs, had *IL-10RA* as a putative target, which was dysregulated in the lesional transcriptome of CCMs [55]. Of interest, lower plasma levels of IL-10 have been reported in CCM subjects who experienced recent bleeding [55].

### 1.5. Conclusions and Future Directions

A complex inflammatory cell environment and several potential immune mechanisms are inherent to CCM (Figure 3). More research on the role of the immune response in CCM is needed and may impact novel therapeutic and biomarker development. Future studies must parse out the individual roles of the inflammatory cells during lesion genesis and maturation and their involvement in clinical sequelae. Single-cell transcriptomic studies of inflammatory cells at various stages of lesion development could lead to more personalized treatments based on the lesion’s developmental stage [91,92,93,94]. These studies could also aid in the identification of molecules that can stratify patients based on lesion severity and prognosticate clinical events. Cytokines and chemokines may serve as biomarkers to predict disease characteristics due to the inflammatory microenvironment and increased vascular permeability. Of interest, IL-1β and IL-10 are currently being tested as biomarkers of hemorrhagic activity in CCM [90]. Other studies will motivate the repurposing of drugs that impact immune cells for therapeutic interventions [15]. In many ways, CCM remains a paradigmatic disease, as these concepts may be applicable to other neurovascular diseases where there is an interplay between vascular dysmorphism, immune response, and inflammation.

## Figures and Tables

**Figure 1 brainsci-13-01336-f001:**
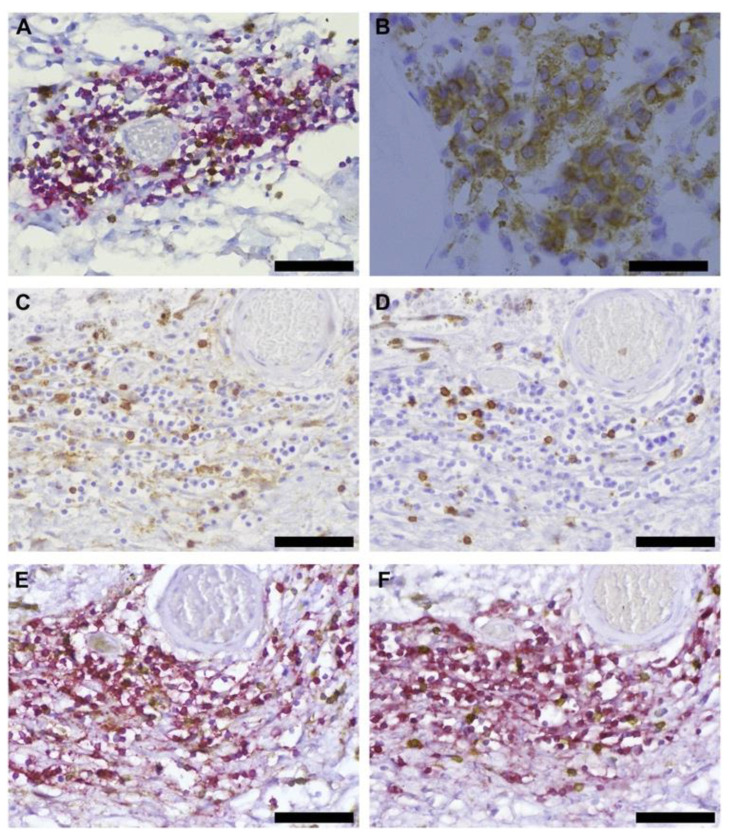
**B and T cells as well as plasma cells in human CCM.** (**A**) CD20^+^ B-cells (red) and CD3^+^ T-cells (brown) were aggregated around the cavern of a CCM lesion. (**B**) CD138^+^ plasma cells (brown) were well-circumscribed around a cavern of a CCM. (**C**) CD4^+^ helper and (**D**) CD8^+^ cytotoxic T-cells were the T-cell subtypes identified in CCM. (**E**) CD4^+^ helper T-cells (brown) and CD20^+^ B-cells (red) or (**F**) cytotoxic CD8^+^ T-cells (brown) and CD20^+^ B-cells (red) were co-localized in CCM lesions, respectively. The original magnification is 400×. All scale bars are 50 μm. (**Figures** were published in *J. Neuroimmunol.*
**272:** Shi, C., et al., Immune complex formation and in situ B-cell clonal expansion in human cerebral cavernous malformations, 67–75 [2014]).

**Figure 2 brainsci-13-01336-f002:**
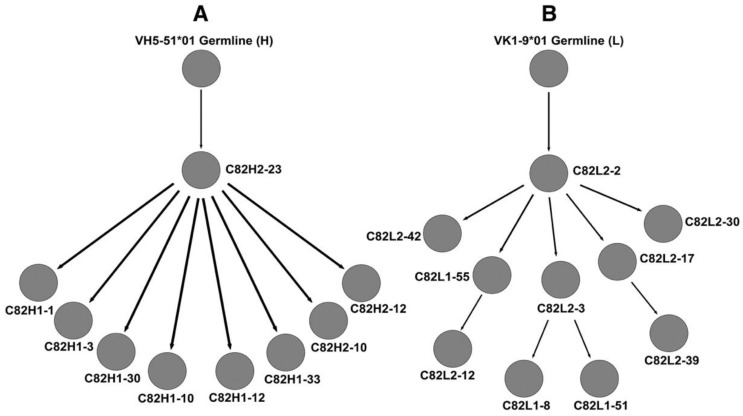
**Clonal tree of B cell in-situ clonal expansion.** Genealogical relationships of germ-line sequences with IgG (**A**) heavy (H) and (**B**) light (L) chain variable regions identified in CCM, with a predicted germ-line clone (**top**). The gradual accumulation and pattern of mutation are consistent with in situ somatic hypermutation. (**Figures** were published in *J. Neuroimmunol*. **272:** Shi, C., et al., Immune complex formation and in situ B-cell clonal expansion in human cerebral cavernous malformations, 67–75 [2014]).

**Figure 3 brainsci-13-01336-f003:**
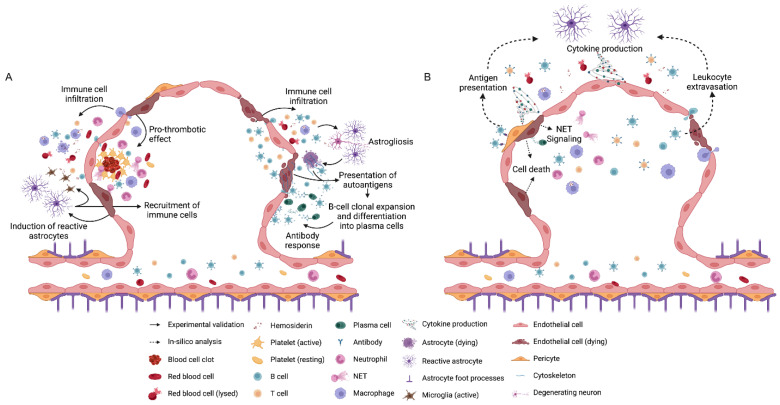
**Diagram of the mechanism of immune response in CCMs.** (**A**) Previous experimental studies showed dysfunctional endothelial cells in CCMs led to extravasation of blood products and leukocyte infiltration. It could create a pro-inflammatory domain, further activating astrocytes and microglia. Cytoskeletal elements of lysed endothelial cells or astrocytes may activate the infiltrated B cells (green star), then cause their clone expansion and differentiation into plasma cells. Activated astrocytes and microglia may release cytokines, which recruit more immune cells to the lesion. In the meantime, the dysfunctional endothelial cells could also show a pro-thrombotic effect and cause thrombi formation and a local hypoxic condition, which could potentially lead to hyperactive angiogenesis. (**B**) *In silico* analyses indicate leukocyte extravasation through leaky endothelium may trigger a vicious cycle of inflammation and damage in endothelial cells. Furthermore, pericytes may serve as non-traditional antigen-presenting cells, which may lead to leukocyte and astrocyte activation and the release of cytokines contributing to endothelial cell death.

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
