# Peer review of "Inflammatory Mechanisms in a Neurovascular Disease: Cerebral Cavernous Malformation"

_brainsci, 2023, doi:10.3390/brainsci13091336_

Round 1

Reviewer 1 Report

Dear Authors,

The manuscript by Li et al. is interesting; however, some improvements are needed. The structure of the review is still not ideal, and the workflow is not clear. Is the review solely describing the diseases, or is it specifically focusing on the adaptive and innate immune systems in these diseases? It is crucial to clarify this aspect since the entire structure of the paper may need to be changed accordingly.

Secondly, there is no clear description of the figures in the manuscript. It is not evident what the authors want to convey, and how these figures are related to the respective sections.

One critical issue arises when the authors inform the reader about the predominance of B Cells. The question that arises is, compared to what? The authors do not show any comparisons in numbers between different sets of immune cells. Additionally, the authors mention clonality in B cells in these diseases without explaining how they arrived at these interesting results. Did they perform sequencing or use other methodologies? Readers would be interested in learning more about this particular issue.

Moreover, the workflow of the manuscript needs to be revisited as the authors keep going back and forth on the same topic in different sections. For example, B cell clonality is discussed multiple times.

The authors refer to CD31 as a marker, but they need to provide more context on its significance in Figure 3. What do they want to show with this marker, and how does it relate to their findings? The manuscript requires more precise explanations throughout.

Regarding future studies, it is not clear what specific aspects the authors intend to explore. Are they focusing solely on inflammation, and if so, what type of inflammation, and which cells are causing it?

At this stage, a broader and deeper scope with precise and clear goals should be highlighted in the introduction.

In conclusion, the manuscript shows promise but requires significant revisions to enhance its clarity and scientific rigor.

Sincerely,

I have not got comments on quality of English language.

Author Response

Reviewer 1. The manuscript by Li et al. is interesting; however, some improvements are needed. The structure of the review is still not ideal, and the workflow is not clear. Is the review solely describing the diseases, or is it specifically focusing on the adaptive and innate immune systems in these diseases? It is crucial to clarify this aspect since the entire structure of the paper may need to be changed accordingly.

            Thank you for your valuable review. First of all, we apologize for the confusion. We restructured the paper with a totally new abstract and a more focused introduction summarizing the aims of the review. We focus now on the innate and adaptive immune systems as specifically studied CCM disease, rather than the previously rambling general comments. We also added a paragraph on how the preclinical studies on the inflammation observed in CCM disease can pave the way for biomarker development with a context of use in hemorrhagic activity of CCM.

Secondly, there is no clear description of the figures in the manuscript. It is not evident what the authors want to convey, and how these figures are related to the respective sections.

            We apologize for the lack of clarity. We modified Figures 1 and 2 (previously Figures 1 and 3) to add more clarity to our manuscript. We also moved Figure 3 (previously Figure 2) into the Conclusion and Future directions section. Finally, we also edited the description of each figure.

One critical issue arises when the authors inform the reader about the predominance of B Cells. The question that arises is, compared to what? The authors do not show any comparisons in numbers between different sets of immune cells. Additionally, the authors mention clonality in B cells in these diseases without explaining how they arrived at these interesting results. Did they perform sequencing or use other methodologies? Readers would be interested in learning more about this particular issue.

            We are sorry for the lack of clarity in these sections. We now present a consolidated section on  “Inflammatory cell infiltrate in CCM, predominance of B cells and their clonal expansion, potential pathogenicity”.  We now discuss the rich repertoire of inflammatory cells, the particular features of B cell clonal expansion in CCM and its implication in terms of antigen honed response and its triggers, and pathogenicity. We then discuss possible roles of other cells.

Moreover, the workflow of the manuscript needs to be revisited as the authors keep going back and forth on the same topic in different sections. For example, B cell clonality is discussed multiple times.

            We edited our manuscript overall for more clarity.

The authors refer to CD31 as a marker, but they need to provide more context on its significance in Figure 3. What do they want to show with this marker, and how does it relate to their findings? The manuscript requires more precise explanations throughout.

            We changed Figure 2 (previously Figure 3). CD31 was only shown previously to highlight the special relationship of the inflammatory dell niches to endothelium. This was too tangential to our message. We now emphasize B cell predominance and clonal expansion in the text and in revised Figures 1 and 2.

Regarding future studies, it is not clear what specific aspects the authors intend to explore. Are they focusing solely on inflammation, and if so, what type of inflammation, and which cells are causing it?

            We modified the introduction to clarify, and changed to section on Conclusion and Future directions accordingly, for better flow and clarity.

At this stage, a broader and deeper scope with precise and clear goals should be highlighted in the introduction.

            We agree, and essentially rewrote the Introduction accordingly.

In conclusion, the manuscript shows promise but requires significant revisions to enhance its clarity and scientific rigor.

Thank you for your comments.

Reviewer 2 Report

A comprehensive, very good quality review, on a relevant and actual topic in its field.

Minor comments

Figure 1: the color code for the immune cells and the significance of red arrows (H, I) should be defined in the figure legend

A synthetic overview of the manuscript in a ‘Conclusions’ section would be most welcomed

Author Response

Reviewer 2. A comprehensive, very good quality review, on a relevant and actual topic in its field.

            Thank you for your reviews.

Minor comments: Figure 1: the color code for the immune cells and the significance of red arrows (H, I) should be defined in the figure legend.

            We modified Figure 1 to add more clarity to this manuscript.

A synthetic overview of the manuscript in a ‘Conclusions’ section would be most welcomed.

            We edited the section Conclusion and Future directions as well as added Figure 3 (previously Figure 2) in that section.

Author Response

Reviewer 3. Overall, the manuscript lacks proper organization and focus on the intended subject matter. While the aim of the manuscript is briefly mentioned in the abstract, this aim is not adequately addressed in the Introduction section. Consequently, comprehending the manuscript's central theme was challenging. Most readers of review papers anticipate the latest information regarding specific topics. Regrettably, this review's description is too simplistic to provide a thorough understanding of the role of inflammatory cells in CCM pathogenesis. Furthermore, the absence of full names for abbreviated terms complicates matters, making it challenging for aspiring researchers seeking to comprehend an unfamiliar disease model and its associated mechanisms through this review.

            Thank you for your valuable reviews. First of all, we are really sorry for our lack of clarity and organization in the first iteration of this manuscript. We re-edited in-depth the flow of our manuscript, as well as write the full names for each abbreviated term.

The primary concern arising from this manuscript pertains to whether the authors have accurately employed the appropriate references throughout. Initially, during the review of this manuscript, I attempted to address these issues individually. Regrettably, I eventually refrained from rectifying the errors due to my waning confidence in the manuscript's accuracy. Presented here are some examples of the necessary corrections:

  • Line 83 (p2): they informed that ‘TNF-1 has been shown to play a role in T-cell adhesion to ECs, extravasation, and activation that occurs during neuroinflammation [32]’. However, there is not such data in ref 33.
  • Line 96 (p3): they stated that “Based of preclinical>….A phase 2 trial targeting ROS to treat symptomatic CCM is ongoing [42]”. However, ref 42 was published in 2015 and is not associated with the mentioned phase trial.
  • Line 152 (p5): They referenced the results from ref 46, suggesting that simvastatin has an impact on B cell depletion and could be repurposed as a potential therapeutic agent. However, the original paper clearly states, "The Pdcd10+/−Trp53−/−/Msh2−/− models showed a mean CCM lesion burden per mouse reduction from 0.0091 in placebos to 0.0042 (P=0.027) by fasudil, and to 0.0047 (P=0.025) by atorvastatin treatment but was not changed significantly by simvastatin." Furthermore, there was no information provided regarding the effect of simvastatin on B cell depletion.
  • In line 183 (page 6), while describing the function of pericytes akin to that of immune-regulating cells, they cited a review paper (#56) instead of the original sources, which should be as follows:

1) Edelman, D.A.; Jiang, Y.; Tyburski, J.G.; Wilson, R.F.; Steffes, C.P. Lipopolysaccharide activation of pericyte’s Toll-like receptor-4 regulates co-culture permeability. Am. J. Surg 2007, 193, 730–735.

2) Kovac, A.; Erickson, M.A.; Banks, W.A. Brain microvascular pericytes are immunoactive in culture: Cytokine, chemokine, nitric oxide, and LRP-1 expression in response to lipopolysaccharide. J. Neuroinflammation 2011, 8, 139:1–139:9.

We agree with the reviewer. We are really sorry for our management of the references that were incorrectly tracked in our bibliography software. We corrected and removed all the incorrect references from this edited manuscript.

Creating a review paper based on existing review papers is, in my opinion, an inadequate approach. While authors can certainly draw insights from other review papers, it is expected that they verify the primary findings and incorporate appropriate references. Furthermore, I would like to highlight that the sentence from lines 183 to 186 (page 6) appears to have been derived from another review paper (ref. 56) which primarily discussed peripheral vascular cells, not brain pericytes.

      We agree with the reviewer. We now cite the original papers reporting the findings rather than review papers.

Another concern arises from their utilization of figures extracted from other published papers. In Figure 1, images were directly replicated from two distinct published sources: one originating from one of the corresponding authors’ groups, and the other seemingly unrelated to the authors. Surprisingly, there is no mention of obtaining approval from the original source, particularly in the case of the unrelated group's content. Furthermore, curiosity has arisen in their depiction of Figure 1J, which shows an immunohistochemical staining of CD34, a marker for stem cells. However, the authors used this image explaining as CD34-positive endothelial cells. To rectify this error, they should have initiated communication with the authors of the original paper to acquire accurate information concerning the CD34 depicted in Figure 1J. Additionally, the manuscript lacks details such as scale bars and reference numbers in the legend, elements essential for a comprehensive understanding of the figures' context. I would like to highlight another case concerning the figures. The images within Figure 3 have been sourced from a previously published paper by this group (below; ref. 20), and without appropriate adjustments, they have attempted to reconstruct a figure.

            We agree with the reviewer. We provide a new Figure 1 to add more clarity to this manuscript. We also acknowledge the sources for each figure as described in the author guideline for this journal.

In addition, we are sorry not noticing earlier this original mistake in previous Figure 3 from our manuscript (Zhang et al., 2020). We will write a corrigendum and edit this figure for publication in the original journal. This image is no longer included herein.

Besides that, there are several other issues that contribute to rendering this manuscript unfavorable. Notably, in line 260 (page 7) of the funding acknowledgment section, they assert that “the NINDS/NIH grant (P01 NS092521) for I.A.A. and D.A.M”. However, it should be noted that D.A.M. was not included in the list of authors.

            We apologize and agree. We edited the funding section (page 13).

Round 2

Reviewer 1 Report

Thank you for addressing my comments.

Minor English checking could be needed.

Author Response

Thank you

Reviewer 3 Report

The authors appropriately addressed the previous comments and improved their manuscript by excluding certain content and introducing a new section entitled 'Biomarkers and Clinical Features of CCM Disease'.

However, a concern I have is that over 25% of the references in the manuscript are self-referential (18 out of 68). Furthermore, certain sections appear to resemble summaries of the authors' own publications, as a significant majority of the references are attributed to the authors themselves. This is particularly noticeable in the first paragraph of the section titled "Inflammatory Cell Infiltrate in CCM, Predominance of B Cells and Their Clonal Expansion, Potential Pathogenicity" (pages 3 to 5), the first and second paragraphs of the section titled "Transcriptomic Studies Highlight Inflammatory and Immune Mechanisms in CCM" (pages 6 to 7), and the section titled "Biomarkers and Clinical Features of CCM Disease" (page 9).

Since the contents of these sections are prepared largely based on the results of the authors' own publication, this review paper might exclusively represent the authors' perspective. If appropriate acknowledgment is given to the work of other researchers, the review papers could attain a more balanced and comprehensive perspective.

In the "Conclusion and Future Directions" section (p11), the authors highlighted the importance of inflammatory cytokines in the context of CCM. Similar phenomena such as elevated levels of cytokines and chemokines, increased ROS production, and immune cell infiltration due to a disrupted blood-brain barrier (BBB) have been observed in various other neuropathological conditions and neurovascular disease models. An important question arises: How can one differentiate CCM-specific biomarkers from those associated with other disease models using the molecules discussed in this review paper?

The final sentence on page 5 reads, "NET formation and release processes also depend on the generation of reactive oxygen species (ROS), a process shown to play an important role in CCM pathogenesis [21]." This sentence could potentially be misinterpreted. While the connection between NET formation and ROS generation is suggested in ref [21], the specific relationship between ROS and CCM was not thoroughly reviewed in that context. Hence, the subsequent comment regarding the Phase 2 trial may not be justified.

Author Response

The authors appropriately addressed the previous comments and improved their manuscript by excluding certain content and introducing a new section entitled 'Biomarkers and Clinical Features of CCM Disease'.

Response: Thank you

However, a concern I have is that over 25% of the references in the manuscript are self-referential (18 out of 68). Furthermore, certain sections appear to resemble summaries of the authors' own publications, as a significant majority of the references are attributed to the authors themselves. This is particularly noticeable in the first paragraph of the section titled "Inflammatory Cell Infiltrate in CCM, Predominance of B Cells and Their Clonal Expansion, Potential Pathogenicity" (pages 3 to 5), the first and second paragraphs of the section titled "Transcriptomic Studies Highlight Inflammatory and Immune Mechanisms in CCM" (pages 6 to 7), and the section titled "Biomarkers and Clinical Features of CCM Disease" (page 9).

Since the contents of these sections are prepared largely based on the results of the authors' own publication, this review paper might exclusively represent the authors' perspective. If appropriate acknowledgment is given to the work of other researchers, the review papers could attain a more balanced and comprehensive perspective.

Response: We would like yo point out that CCM is a rare disease, and our group has been a leading contributor to its literature, and the only one to have published controlled analysis of inflmammatory cell infiltrate in CCM in comparison to other cerebrovascular anomalies and studies of clonal expansion of B cells in lesions, and their antigenic trigger. Our team is the first or only to have published detailed lesion transcriptomics, transcriptomics of lesion at early and late stages of disease, and the only one to have published biomarkers of hemorrhage in the disease, and any prognostic biomarkers. Where appropriate, we cite other publications, like GWOS analyses of pro inflammatory genes.

Our group has received the most NIH funding for CCM research in the past 20 years. A recent bibliometric analysis of the most cited CCM related articles by Fry, et al. (WORLD NEUROSURGERY, https://doi.org/10.1016/j.wneu.2022.11.042) included 25 papers with co-authors by our team among the 100 most cited papers in the field.

The Special Issue Editor who invited us to submit this review specifically did so because of our team's contributions.

In the "Conclusion and Future Directions" section (p11), the authors highlighted the importance of inflammatory cytokines in the context of CCM. Similar phenomena such as elevated levels of cytokines and chemokines, increased ROS production, and immune cell infiltration due to a disrupted blood-brain barrier (BBB) have been observed in various other neuropathological conditions and neurovascular disease models. An important question arises: How can one differentiate CCM-specific biomarkers from those associated with other disease models using the molecules discussed in this review paper?

Response: There is no suggestion that biomarkers are specific to CCM. We now mention this in the conclusion, adding on page 9" "While many of the biomarkers are not specific to CCM, and have been invoked in other inflammatory conditions, they have been linked to the CCM lesion transcriptome and CCM-specific contexts of use".

The final sentence on page 5 reads, "NET formation and release processes also depend on the generation of reactive oxygen species (ROS), a process shown to play an important role in CCM pathogenesis [21]." This sentence could potentially be misinterpreted. While the connection between NET formation and ROS generation is suggested in ref [21], the specific relationship between ROS and CCM was not thoroughly reviewed in that context. Hence, the subsequent comment regarding the Phase 2 trial may not be justified.

Response: We add a qualifier to the sentence on page 6 "While the relationship of NOS to CCM pathogenicity requires further study,.." when mentioning the Phase 2 trial.